# EMG-Centered Multisensory Based Technologies for Pattern Recognition in Rehabilitation: State of the Art and Challenges

**DOI:** 10.3390/bios10080085

**Published:** 2020-07-26

**Authors:** Chaoming Fang, Bowei He, Yixuan Wang, Jin Cao, Shuo Gao

**Affiliations:** 1School of Instrumentation and Optoelectronic Engineering, Beihang University, Beijing 100083, China; timefliesfang@buaa.edu.cn (C.F.); wangyixuan@buaa.edu.cn (Y.W.); 2School of Automation Science and Electrical Engineering, Beihang University, Beijing 100083, China; 17374556@buaa.edu.cn; 3Department of Psychiatry, Massachusetts General Hospital, Harvard Medical School, Boston, MA 02138, USA; jcao9@mgh.harvard.edu; 4Beijing Advanced Innovation Center for Big Data-Based Precision Medicine, Beihang University, Beijing 100083, China

**Keywords:** multisensory, electromyography, pattern recognition, rehabilitation

## Abstract

In the field of rehabilitation, the electromyography (EMG) signal plays an important role in interpreting patients’ intentions and physical conditions. Nevertheless, utilizing merely the EMG signal suffers from difficulty in recognizing slight body movements, and the detection accuracy is strongly influenced by environmental factors. To address the above issues, multisensory integration-based EMG pattern recognition (PR) techniques have been developed in recent years, and fruitful results have been demonstrated in diverse rehabilitation scenarios, such as achieving high locomotion detection and prosthesis control accuracy. Owing to the importance and rapid development of the EMG centered multisensory fusion technologies in rehabilitation, this paper reviews both theories and applications in this emerging field. The principle of EMG signal generation and the current pattern recognition process are explained in detail, including signal preprocessing, feature extraction, classification algorithms, etc. Mechanisms of collaborations between two important multisensory fusion strategies (kinetic and kinematics) and EMG information are thoroughly explained; corresponding applications are studied, and the pros and cons are discussed. Finally, the main challenges in EMG centered multisensory pattern recognition are discussed, and a future research direction of this area is prospected.

## 1. Introduction

Monitoring and analyzing the patient’s physiological information are of significance in the process of physical rehabilitation in order to evaluate the rehabilitation effect [1,2] and control auxiliary devices during the physical rehabilitation process. Conventionally, the physiological information is divided into physical and psychological information, e.g., muscle force information and the intention of the patient. To detect these two categories of information, various types of sensors, including electromechanical sensors (such as accelerometers [3,4], gyroscopes [5,6] and force sensors [7,8]), and biosensors (such as electromyography (EMG) [9,10,11] magnetoencephalography (MEG) and electroencephalogram (EEG)) have been utilized. Electromechanical sensors are capable of detecting the physical information effectively. For example, in [12], flex sensors and force sensitive sensors (FSRs) are utilized to detect the bending angle of the finger and the grasp force, respectively. The sensing information is then used for the control of robotic fingers in performing ten different tasks. In [13], the integrated use of accelerometer sensors, pulse sensors, and ambient sensors is reported for recognizing sedentary behavior with an accuracy of 95%. Nevertheless, biosensors are not only proven to reflect physical information, but convert the bioelectric signal, which is broadly treated as a direct link to human psychological information, into interpretable voltage amplitudes, and therefore have unique advantages in psychological detection over the electromechanical sensors [11,14]. Note that in the rehabilitation area, human psychological information normally refers to human intention information, which can be used to examine the synchronization level between human movements and human thinking. Hence, in this article, the detection of human intention, instead of human psychology, is focused.

Among all biosensor-captured information, EEG, MEG, and EMG are the three most relevant signals to human intention [15,16,17]. Among them, the EEG signal is of weak robustness due to the shortage of noninvasive electrodes in collecting a surface EEG signal, failing to provide high signal quality. MEG techniques, e.g., MRI, can offer accurate and rich information, but the time-consuming issue and huge machine volume block its use in rehabilitation. In contrast, EMG techniques enjoy relatively higher signal-noise ratio (SNR) and robustness than EEG means (especially during movements), and process information much faster than MEG techniques. Therefore, EMG is a more preferred choice for intention detection in the rehabilitation field.

Two widely used techniques to interpret the EMG signal are signal intensity registration and pattern recognition [18,19]. The former approach is based on the correlation between EMG signal intensity and quantified muscle force level [20,21]. However, the EMG amplitude of a muscle is usually determined by many uncontrollable factors, e.g., EMG crosstalk [22] and variation in muscle force [23], which will not only generate misinterpretations of the muscle activities, but also result in potential safety risks when such information is directly utilized in the exoskeleton for the motion functions impaired patients [18]. Alternatively, the latter approach offers higher accuracy in decoding EMG signals owing to the higher level of information, e.g., a set of motions or the movement intention can be extracted via raw data to provide deep insights in examining user’s body condition. EMG PR was not popularized for its high computational cost compared to its counterpart, until being promoted by the rapid development of electronics and information technology in recent years. Now EMG PR has been broadly applied to disease diagnosis [24], intelligent prosthesis control [19], and gains continuing attentions from relevant researchers.

Generally, the recognition scenarios can be divided into upper limb PR and lower limb PR. In early years, research on EMG PR mainly focused on the recognition of the upper limb movements with great differences, e.g., largely changed arm movements [25]. This is because the EMG patterns of such movements are significantly different and therefore is easier to be recognized. Nevertheless, fine upper movements and lower limb movements are also important in many neuromuscular disease analysis, such as Parkinson’s disease and myasthenia. Therefore, it is also important to create a reliable relationship between EMG signal and patients’ intentions and physical conditions for these two kinds of movements.

However, using the EMG signal to interpret fine upper limb and lower limb motions is challenging. In terms of the former, small movements in the upper limbs like the finger or wrist only result in slight differences in EMG signals, giving rise to difficulties in effectively distinguishing the EMG signal of one pattern from others. For example, a hand movement recognition method using single-channel sEMG is presented in [26]. This work reached an accuracy of 86.7% in classifying nine finger movements, and the recognition accuracy and the pattern types are all lower than the multisensory approach [27]. As to the latter, interactions between lower limbs and unpredictable environments during movements can generate unexpected EMG patterns. For example, the muscle activity of lower limbs is often related to the flatness of terrains [28], or the different locomotion modes such as obstacle crossing [29] and stairs ascending/descending [30]. Therefore, utilizing only the EMG signal cannot satisfy the demand for reaching robust recognition performances at these complicated scenarios without learning environmental knowledge. A possible solution is to concurrently acquire the external information of the environment as well as the EMG signals, so that information from different dimensions can be integrated for further analysis and complementing each other.

In light of this, EMG-centered multisensory-based technologies emerge in recent years, and increasing demonstrations and attempts in three main rehabilitation scenarios have been reported globally. First, multisensory fusion EMG PR plays an important role in accurate and early diagnosis for neuromuscular diseases so that the patients could start the rehabilitation process before the diseases further develop. For example, in [31], the acceleration information of the muscles and forearm of the biceps brachii of the Parkinson’s patient are collected simultaneously, 12 kinds of signal features were extracted for cluster analysis, and successfully achieved a diagnostic accuracy rate of more than 90%. Second, multi-sensor fusion can evaluate the rehabilitation process of patients and the effectiveness of the treatment in more dimensions. For example, in [32], the fusion analysis of the plantar force information of the Parkinson’s patient and the myoelectric information of the anterior tibial muscle. The effect of a plantar sensory stimulation therapy on improving gait and motor output in Parkinson’s patients were evaluated. Third, multisensory EMG PR is used as the control strategy for assistive devices such as intelligent prosthesis and exoskeleton [33,34,35,36], these auxiliary devices can effectively help patients in physical rehabilitation, and to a certain extent reduce the burden of the physical therapy. In addition, multi-sensor fusion also allows these devices to obtain higher security and a more comfortable user experience. For example, in [37], an exoskeleton hand is designed to help paralyzed people in rehabilitation training. It uses myoelectricity as an input to represent the patient’s intention and uses angle information and gripping force information as feedback to provide reference and correction for EMG signals in real-time, ensuring that the rehabilitation action can be completed accurately. The above three application scenarios are conceptually depicted in Figure 1.

To provide a timely and systematic overview of the EMG based pattern recognition techniques using multisensory fusion strategies in the context of rehabilitation, this review article is composed. The paper is structured as follows: we first introduce the basic physiology knowledge relevant to EMG-based PR in Section 2. Then, in Section 3, a detailed explanation of the processing procedure and comparison of the state-of-the-art EMG PR is presented. Afterward, Section 4 explains the collaborations between EMG and multi-sensory information and presents diverse multisensory strategies successfully applied in different rehabilitation scenarios. Finally, the challenges faced by EMG PR and the prospect in resolving these challenges are summarized in Section 5.

## 2. Physiology Background

For reader convenience in understanding the working principle for EMG pattern recognition, some relevant physiological backgrounds, including the mechanism of EMG generation and the target patterns, are essential prior knowledge. This chapter will explain these physiological backgrounds, which helps readers to understand the characteristics of EMG signals and the design of recognition systems for the target patterns.

### 2.1. EMG Signal Overview

EMG is a type of technique used to evaluate and record a series of electrical signals that emanate from body muscles [38]. The principle of the generation of the EMG signal can be described as follows. Motor nerve cells produce electrical pulses under the control of the central nervous system in the cerebral cortex. These neural signals are transmitted to muscle fibers through axons and cause pulse sequences, which activate them to contract and produce muscle tension. Meanwhile, a current is generated in the human body that brings about transmembrane potential [39]. The transmembrane potential is the difference between the internal and external potentials of muscle cell membranes. When muscle cells are in a quiet state, the cell membrane potential is polarized. The potential difference between the inside and outside of the cell membrane under the polarization state is called resetting potential. Depolarization occurs when the cell is excited, and this trend will spread to around [40]. The corresponding action potential is defined as an electromyography signal. The above procedure is demonstrated in Figure 2.

### 2.2. Human Movement Patterns

As the EMG signal is a bioelectrical signal controlled by the mutual effect of the receptors and nerve system of human beings, the pattern of the EMG signal relies on both the user’s subjective intention and the interactive environment conditions while performing specific actions. In this section, we will introduce some frequently studied upper and lower limb patterns in relevant studies.

The upper limbs of the human body include a forearm, elbow, rear arm, wrist, and hand. The muscle mass of the upper limb is generally small and slender, and the movement range of the upper limb is not extensive. Compared with the lower limbs, the muscle strength of the upper limbs is generally weaker because the body does not need to be supported.

The main functions of the upper limbs are grasping, stretching, and expressing information with hand gestures. Therefore, the patterns in the upper limb can be divided into two categories: limb position and hand gestures. The classification of Limb positions has been elaborated comprehensively in [25], including P1: arm hanging at the side, elbow bent at 90 degrees; P2: straight arm reaching up (45 degrees from vertical); P3: straight arm hanging at side; P4: straight arm reaching forward. Hand gestures account for the dexterous movements of the hand, including wrist and finger movements. Wrist gestures are the hand movements that rotate the whole hand around the wrist joint, but finger gestures only involve the movements of the fingers. Reference [27] proposed a challenging set of 15 gestures in which 5 (thumb, index, middle, ring, pinky) are finger gestures, 6 are wrist gestures. The above patterns are demonstrated in Figure 3.

The lower limbs of the human body include the hip, thigh, calf, foot, hip, knee, ankle, and other joint parts. The lower limb has the characteristics of larger muscle tissue and a larger joint movement angle [41]. Its function is mainly to support the human body and change the position of the human body. Depending on these physiological functions of the lower limbs, the activities most closely related to the lower limbs are often activities with the core of walking behavior, so the activity recognition based on the lower limbs can be roughly summarized in two aspects, namely, the locomotion mode and the gait phase.

The first one is locomotion mode, which aims at the process that the brain actively coordinates the limbs to make different actions when the human body is facing different environments in the real world. Specifically, up and downslope, walking on flat ground, up and downstairs are the most widely studied in the relevant research [35,42,43]. This is not only because these five sports modes are the most common and widespread scenes in life, but also because these modes have the process of overcoming the influence of gravity and changing the center of gravity of the body, so auxiliary equipment is needed to adjust the lower limbs [44,45]. In addition to these five kinds of sports modes, some work has also been done to study such sports modes as crossing obstacles [29,46], turning [29], standing, and sitting transition [47]. These locomotion modes are a more in-depth interpretation than simple actions, such as leg swinging, touchdown, etc., which is conducive to the auxiliary equipment to have a better understanding of the user’s environment and generate adaptive control based on this.

Gait phase is another widely concerned lower limb mode, which is a scientific decomposition and utilization of human periodic walking movement. There are different ways to define the gait phases according to different usage scenarios. The simplest one is to divide the gait into two phases: namely, the stance phase and swing phase [3]. Generally, the heel strike (HC) and toe-off (TO) are used as the starting point of the stance phase and the swing phase, respectively [48]. However, this division is not accurate enough and is not applicable in scenes that require high continuity. In [42], a definition of the gait phase consisting of four phases is proposed and used for real-time recognition. In the most precise definitions, the gait cycle can be divided into eight phases [49].

The function of the gait phase is to provide auxiliary equipment to refine the human walking process and provide different auxiliary control in different gait segments [42,50]. The characteristics of the gait phase, such as continuity, duration, and so on, to describe the whole walking cycle of the human body, is a standard method in gait analysis and has been proved to be related to the diagnosis of some diseases [51]. Figure 4 shows the five most commonly studied locomotion modes, as well as a detailed definition of the gait phases.

## 3. EMG Pattern Recognition Pipeline

The differences in human body motion patterns can be reflected in EMG signals. However, the raw EMG data is usually very noisy and cannot be directly classified. Therefore, a series of processing procedures is necessary to distinguish these differences accurately. Generally, the PR procedure can be summarized in the following four steps (also depicted in Figure 5):Data acquisitionSignal preprocessingFeature extraction and reductionClassification

There is no universal standard for these processing procedures yet, owing to the variability for EMG PR in different rehabilitation scenarios. Therefore, in this section, we will explain the EMG PR protocols and approaches adopted by relevant studies in detail.

### 3.1. Data Acquisition

Acquiring a high-quality EMG signal is strongly desired to ensure a satisfying recognition accuracy level, so properly designing the front-end EMG acquisition approaches is very important. In this section, we will compare the methods in front-end EMG acquisition from two aspects: the design of the EMG sensing system and the selection of muscle measurement points.

#### 3.1.1. EMG Sensing System

The sensing system for collecting EMG signals is mainly composed of electrodes, amplifiers, microprocessors, and transmission devices. The electric signal generated by the muscle is picked up by the electrodes, amplified by the amplification circuit, and then transmitted to the host computer via the transmission device. Generally, the connection between electrodes and processor is wired while the transmission is typically wireless.

In the collection of EMG, the design of the sensor module is typically highly similar, except for the electrode. Common EMG electrodes include two main kinds of pole placement methods: monopolar electrode and bipolar electrode. Both placing methods measure the potential with reference to the electrode placed at the locations without EMG response (e.g., ankle or knee), while the Monopoles method directly measures the potential difference and bipolar method apply the differentiate amplification method [52]. Bipolar electrodes have a higher usage frequency because the common mode noise can be suppressed in real application but lack of setup flexibility when compared to monopoles. Besides, the EMG signal quality is also influenced by the distance between each electrode pole and their diameters and widths when bipolar electrodes are applied [38,52].

In addition to electrode placement, the implantation of the electrode is also reviewed here. Intramuscular EMG and surface EMG are two mainstream Electromyography signal acquisition schemes. Usually, intramuscular EMG signals are recorded using percutaneous fine wire electrodes or others made of similar materials. These electrodes need to be inserted into several muscle tissues using hypodermic needles [53]. Insertion locations are identified by palpation [54] and verified by electrical stimulation and EMG channel activity during corresponding test contractions. Different from intramuscular EMG signal collection, surface EMG (sEMG) electric potentials are acquired with electrodes placed on the skin just above the target muscle [52]. Electrodes occurring in previous experiments are usually made of silver. Typically, there are two ways for operators to contact the electrode with the skin. One is by using a silver-chloride gel to achieve wet contact while the other applies dry electrodes with microneedles to record EMG signals [43].

According to the literature research, surface EMG hosts a higher usage frequency compared with intramuscular EMG [52,55,56,57]. The main reasons for its wide application are regarded to be non-invasive and convenient. Modern surface EMG electrodes are stuck to the surface of muscle and avoid the danger of causing potential muscle injuries to tissue, which is considered to be the critical concern in intramuscular EMG. Besides, during the use of sEMG, operators can freely select measurement points on the surface of muscle tissue and reduce the time cost to do the preparation work like disinfecting the probe points with alcohol. Nevertheless, some bold attempts in the application of intramuscular EMG have also been provided recently [53,54]. Intramuscular EMG provides another signal source to detect the human body behavior and handles some difficulties related to sEMG-based control, such as the unstable contact of the electrode with the skin. Some other additional benefits of intramuscular EMG are given in [53], for example, the ability to record deep muscle signals with little EMG crosstalk. Although the experimental results in [53,54,58] all prove that intramuscular EMG could not bring with the reduction in classification error from sEMG for single classifiers while the significant decrease of classification error for parallel classifiers can be achieved. According to the practical efforts in [53,54], the parallel configuration for simultaneous control becomes more promising via the use of intramuscular EMG, which has long been an unsolved problem in sEMG studies.

#### 3.1.2. Muscle Site Selection

In the thesis of movement recognition, reasonable muscle group selection for EMG signal obtainment is of considerable significance for robust recognition. Various muscle groups in both the lower and upper limbs are functioning and positioned differently, and Figure 6 demonstrates the anatomical structure of the muscles.

Real muscle selection needs to consider the previous experience and specific demand in the application scenario. For example, as for prosthetic control, which is mainly designed for the amputees, experimenters commonly choose the proximal muscle groups like thigh muscle group [29,35,42] and the upper arms [59] because the distal ones are considered to be amputated. While, as for the studies oriented for non-amputees subjects, the distal muscle groups such as the shank muscle tissues and forearm and wrist muscles are often selected as targeted muscles due to their lower inter-subject variability than the proximal [43,60,61]. Gao Shuo [43] chose a pair of distal antagonistic muscles, tibialis anterior (TA) and soleus (SL) to conduct the terrain identification experiment and got satisfying results. Besides, muscle characteristic is also an important aspect to be taken into account. For example, the proximal hip muscle groups (AM, GM, and PRF) are taken into consideration in [9] due to their sensitivity to different walking speeds. It has been proved in [61] that rectus femoris (RF) and Soleus (SL) are responsible for pulling the body forward at the stance phase of the gait and therefore studies related to the recognition of uphill/upstairs behaviors demanding to overcome the gravity are more likely to choose such muscles for EMG detection [62]. The selected criterion and the specific muscle groups selected by relevant studies have been summarized in Table 1.

### 3.2. Signal Preprocessing

The original EMG signals obtained through the front-end acquisition equipment are generally noisy, and pattern recognition requires the use of signals of limited length for further processing. Therefore, this section will introduce filtering and windowing preprocessing methods to reduce the noise in raw data and segmenting the EMG signals.

Real EMG signals are usually mixed with various kinds of noise due to environmental disturbance and other uncontrollable factors in the experiments. Reasonable preprocessing is necessary for extracting useful features in further analysis. Filtering, normalization, and windowing are the three most common EMG data preprocessing methods. The filtering technique derives from classical signal process schemes of signal preprocessing in the frequency domain. Most of the energy of the EEG signal are mainly within the frequency range 0–500 HZ [65]. In [66,67], it is stated that high-frequency (500–1000 HZ) EMG signal is likely to be interfered with by the aliasing. As for the low-frequency region (1–10 HZ), it is mainly regarded as a kind of noise caused by the cable movement and the interface of the measurement electrode and the skin. Actually, the most commonly used EMG signal filter is essentially a finite impulse response bandpass filter with a cutoff point of 10 HZ (low) and 500 HZ (high) [68,69,70].

Due to the difficulty when dealing with the long-term sensor data sequence, windowing operation is necessary to slice it into short clips for the real-time prediction task. Generally, the properties of the signal decide the window length when analyzing the EMG data. According to [71], EMG signal can be considered a wide-sense Gaussian random process. Therefore, proper window length is of great significance for information extraction. The too-long window will lead to clips with an unbearable variance, while too shot sequence may not be able to contain enough useful information for classification [72]. Huang, He et al. [29] state that the analysis window length should not exceed 200 ms, which is an ideal upper bound to control the signal variation. Research in [8,72] suggests that windows of 150–250 ms for EMG are the optimal tradeoff between the classification accuracy and the delay, while 100–250 ms windows can be used for mechanical sensors. Fast time response is also desired for continuous classification tasks in addition to high classification accuracy [29]. In [42,72], a method called overlapping analysis windows was applied to accelerate the decision update. The key aspect of this scheme lies in the careful window increment setting. The smaller the window increment step, the faster the classification result can be achieved. Furthermore, overlapping can help obtain better classification performance when processing the transition phase, such as from the sitting phase to the standing phase [42].

### 3.3. Feature Extraction

After the signal was preprocessed, the noise was attenuated and the window segments were obtained. However, using the entire segment as the input data is not an ideal choice due to the high computation cost and poor correlation between the input data and the target patterns, which would negatively affect the recognition performance. Therefore, it is necessary to extract some features from each window sequence. In this section, the two most commonly applied feature extraction approaches, including time-domain features, frequency domain features, and autoregressive features, are reviewed and compared.

The time-domain features of the EMG signal are based on the characteristic index of the statistical method, which regards the EMG signal as a function of time. This is an intuitive interpretation of the EMG signal. Its calculation is directly related to the magnitude of window segments, which occupies small calculation resources. In [71,73,74], Hudgins et al. proposed several time-domain features that are widely used in the following researches, including mean absolute value (MAV), mean absolute value slope, slope sign changes (SSC), waveform lengths (WL) and zero crossings (ZC). In [75], time-domain features like root mean square (RMS), waveform length (WL), number of zero-crossings(ZC), variance (VAR) and maximum(MAX) value are combined to represent the feature space.

The frequency-domain features of EMG signal are mainly to transform the time-domain signal of EMG signal into a frequency-domain signal through Fourier transform, and then analyze the spectrum characteristics or power spectrum characteristics of the signal. The advantage of this method is that it overcomes the characteristics of a time-domain signal, which is greatly affected by noise, is relatively weak, and it is easier to extract stable characteristic indexes by analyzing EMG signals in frequency-domain. Common frequency domain features are comprehensively investigated in [76]. Lots of experimental results demonstrate that mean frequency (MNF), median frequency (MDF), mean peak frequency (PKF), mean power (MNP), frequency ratio (FR), power spectrum ratio (PSR), and variance of central frequency (VCF) are not good features for EMG signal-based locomotion mode classification. Joint time-frequency domain features have been proved to be capable of effectively representing transient EMG patterns resulting from dynamic contractions [77]. The experiments comparing the time domain methods and time-frequency methods have also been conducted, and results show that the wavelet packet transform (WPT) is the best method to increase the EMG information density.

Apart from the conventional time and frequency domain feature extraction approaches, there are other ways of extracting effective features. For example, autoregressive (AR) features are usually taken into consideration in lower limb locomotion mode analysis. Huang. He et al., applied autoregression features (three-order autoregression coefficients) and time-domain features (MAV, ZC, and WL) for locomotion identification for their computation efficiency and fast time-response property.

In recent years, there are also handcrafted features that do not rely on statistical techniques. Instead, they use the prior knowledge of biomechanics to select the specific response point at the experimental curve. For example, in [43], twenty-one biomechanical features such as the peak EMG interval between two selected muscles and the duration of the EMG activation time are carefully selected. Such an extraction technique obtained an accuracy of 96.8% and is proven to outperform the traditional feature extraction technique. Although this kind of method seems lacking the general capability dealing with different scenarios, the classification performance is satisfying for the only specific task and shows less generality than the conventional features. There are also researches dedicating to improving the efficiency of the conventional feature extraction methods by designing new features. For example, in a most recent research [78], three time-domain features, including ASS, MSR, and ASM, are specially proposed to improve the performance of EMG-PR based strategy in arm movement classification task. Experimental results showed that the new-designed features could achieve the accuracy 92.00% ± 3.11%, which is 6.49% higher than that of commonly used time-domain features [72].

### 3.4. Dimensionality Reduction

When the extracted feature set occupies a high-dimensional space, it may lead to the failure to correctly classify the patterns because the discriminative features are hidden in the complex feature space. Efficient data dimensionality reduction technique not only reduces the computation cost but also proves to be powerful in increasing the inter-class distance, enhancing recognition accuracy. Two commonly used dimensionality methods, i.e., feature projection and feature selection, are being discussed in this section.

#### 3.4.1. Feature Projection

The feature projection method means projecting the original high dimension feature space into a lower-dimensional feature space. The most important characteristic of such a method is that it uses all the original features to get the reduced feature space, which will not bring any information loss. Many algorithms are proposed to accomplish this task, including approaches like principal components analysis (PCA) [77,79], Non-negative matrix factorization (NMF) [80], averaging [75], independent components analysis [81], nonlinear projection [82] and. We will mainly introduce the PCA and NMF approaches in this section.

PCA is regarded as one of the most commonly used linear dimensionality reduction techniques. It essentially takes the direction with the largest variance as the main feature and off-correlates the data in each orthogonal direction, which makes them irrelevant in different orthogonal directions. Therefore, PCA also has some limitations. For example, it can remove linear correlation very well, but there is no way for higher-order correlation. For data with high-order correlation, Kernel PCA can be considered, and the nonlinear correlation can be achieved through the Kernel function turning to linear correlation. The research in [83] illustrates the importance of having relevant embedded muscle activity features in a low-dimensional space. It is also demonstrated in this paper that the PCA technique can efficiently capture features from EMG signals in an unsupervised manner. Due to the parameterless characteristic, PCA is convenient for universal implementation but having trouble for personalized optimization itself. Besides, some other methods about the strategic combination of features have also been leveraged in recent papers, and they are improving towards the direction with higher speed and accuracy. For example, FastICA is proposed in [84] to overcome the difficulty of previously used ICA for high-density EMG signal decomposition.

Non-negative Matrix Factorization (NMF) is another dimensionality reduction technique that has also been widely studied in recent years [80,85]. D.D.Lee et al. point out in [85] that NMF reduces data dimensionality by approximately factorizing the source data into two matrices. The factorization aims to extract the latent structure from the source data to a lower-dimensional space by minimizing the predefined cost function whose value is inversely proportional to the quality of the approximation. Common cost functions include conventional least square error and Kullback-Leibler divergence. As for its application in EMG signal recognition, G.R.Naik and H.T.Nguyen first studied the identification of EMG finger movement with a nonnegative matrix factorization technique [80]. Recently, non-negative matrix factorization is also leveraged to select the subject-specific signal channel for improving the lower-limb motor imagery recognition accuracy. Apart from this, NMF is also found to be put into practice in the determination of muscle synergies while utilizing EMG signals [86].

Actually, lots of other data dimensionality reduction methods have been investigated in recent years due to the quick progress in data science, like Uncorrelated Linear Discriminant Analysis (ULDA) [87], the locality preserving projections (LPP), neighborhood preserving embedding (NPE), discriminant analysis (OFNDA) and so on. These methods provide different dimensions to increase the distinguishable information density, which will help improve recognition accuracy and processing speed.

#### 3.4.2. Feature Selection

Feature selection means reasonably selecting a subset of features to form the new feature space, while the rest of the original features would be abandoned [88,89]. It has been proved by numerous empirical results that feature reduction can reduce data dimensionality and the computational complexity meanwhile.

A key problem in the feature selection method is that what factors should be taken into consideration to define the best EMG feature space, Zardoshti-Kermani, Mahyar, et al. [90] came up with some standards to evaluate the EMG feature space including maximum class separability, robustness, and computational complexity.

Diverse approaches have been proposed to help select more representing features from the dataset containing too much useless data. Some work derives from combining the specific physical or statistical properties. In the task like limb posture classification, deterministic approaches have been proposed to select the optimal feature-channel pairs [91]. In this work, a distance-based feature selection to determine a separability index is utilized. Besides, to measure the amount of mutual information between features and classes, a correlation-based feature selection method was applied, and both schemes were proved effective to boost the posture classification accuracy.

Evaluating relevant features is another novel approach to make feature selection proposed in recent years [92,93,94,95,96,97]. One newest work formulates the feature selection problem as the optimization problem and proposes a personal best guide binary particle swarm optimization (PBPSO) to solve the feature selection, which works by evaluating the most informative features from the original feature set [93]. In another recent paper [97], a feature selection algorithm called ReliefF is given from the heuristic angle. This algorithm first selects the random instance of one of the database classes and then searches for the k nearest neighbor instances.

### 3.5. Classification Algorithms

Once the features with the reduced dimensionality are determined, classification algorithms should be deployed to distinguish the different categories of the extracted feature vectors. Numerous algorithms have been proposed, and in this section, we will review some of the most frequently utilized algorithms in EMG PR, along with the comparison between them.

Traditional classification methods include linear discriminant analysis (LDA) [98], support vector machine (SVM) [99,100], k nearest neighbor (KNN) [101,102], Bayesian analysis, fuzzy logic (FL) [103] and hidden Markov models(HMM) [104] while some modern methods have also been given like artificial neural network (ANN) [105] and convolutional neural network (CNN) along with the recent progress in deep learning research. All of the above methods are effective in classifying the extracted feature space while they each have specific characteristics due to their intrinsic difference at the algorithm level. Therefore, we will briefly explain the basic principle of each type of algorithm along with the comparison of their classification performance at different rehabilitation scenarios.

LDA is one of the most simple but effective methods which has attracted great attention in recent years. Linear discriminant analysis is utilized in for lower limb prosthesis movement mode recognition and achieves competitive accuracy of 97.45% even compared with neural network classifiers while deploying a PCA reduced feature set. In another recent work [105], LDA in a One-Vs-One topology was used for non-weight bearing lower-limb movement recognition immediately after training. Nevertheless, a fatal disadvantage for the LDA model is that it is not capable of handling the linear inseparable problem even if given perfect data. At this point, ANN enjoys the ability to describe nonlinear class boundaries among different categories. MLP and Cascade are the two main structures of artificial neural networks. Identification and classification of different gait phases according to the collected EMG data is a classical linear inseparable problem. In new research focusing on this problem [104], three different MLP is deployed for the classification of two main gait phases while directly using the raw EMG data without feature extraction. The experimental comparison demonstrates that the performance of three MLP models achieves an average higher accuracy over all the popular models. On the other hand, in scenarios that some dimensionality reduction techniques like PCA have linearized the discrimination task, LDA can behave even better than the ANN methods. Besides, due to the well-defined classifier inner architecture, LDA possesses better performance stability. Nevertheless, for a given problem, it may be not easy to determine the optimal size and structure of an ANN. At last, in spite of the advantage of extracting latent features from unprocessed EMG data, the long training period is another thing that we need to take into consideration when intending to apply the ANN.

The SVM is estimated as one of the most popular approaches utilized in the movement mode classification. It is an effective data classification approach that projects the low-dimension data to the high-dimension feature space via kernel function. SVM works by finding a hyperplane to distinguish different categories in high-dimension space through the training process. It is believed that proper kernel function can help reduce the indivisible linear data to linear separable set in high-dimension space. From this, it is obvious that the most crucial problem when building the SVM model is to determine the optimal kernel function and its parameter values. Pires, Ricardo, et al. [99] uses the SVM algorithm to classify lower-extremity EMG signals during running shod/unshod with different foot strike patterns. It is also mentioned that kernel function selection, and parameter setting should depend on the specific task scenario. In [100], the SVM algorithm is utilized to develop an automatic classification system for lower limb hemiparetic patients. The empirical results demonstrate that SVM has the highest accuracy (95.2%) than KNN (89.2%) and ANN (92.3%). Particle swarm optimization (PSO) has long been a classical heuristic optimization algorithm. According to our literature review, the related works hybridizing the PSO and SVM to detect movement patterns are growing continuously in recent years. Zheng, Jiajia, et al. [106] employ PSO-optimized SVM to classify four gait phases, including initial contact, mid stance, terminal stance, and swing phase. The experimental results show that PSO-SVM exhibits the distinctive advantages on gait phase classification and improves the classification accuracy up to 32.9%–42.8% compared with the classifier based on vanilla SVM. Recently, some work compares the performance of common classifiers in different scenarios to provide references in practical application. The comparative analysis among NLR, MLP and SVM shows that for either classification performance and for the number of classification parameters, SVM attains the highest values followed by MLP, and then by NLR [107]. It should also be noted that SVM can obtain the highest classification performance while utilizing the lowest sampling rate.

Fuzzy Logic (FL) is another technique used in the classification of EMG signals, which achieves definite conclusions just from imprecise data input in a simple manner. It has been studied that FL exhibits the advantage of control techniques in biosignal processing [103]. FL can extract unrepeatable EMG features and mimic user’s intent to make a decision. On the other hand, FL requires more system memory and processing time because of the use of fixed geometric-shaped membership functions in fuzzy logic limits system knowledge more in the rule base than in the membership function base [103].

The application of the hidden Markov model (HMM) is described in [108], where it is used for recognition of gait mode based on electromyographic signals. A modified Baum-Welch was used to estimate the parameter of HMM, and the Viterbi algorithm achieved the recognition of gait mode by finding the best HMM and state to assign corresponding phases to the given segments.

KNN classifier is another kind of common biosignal classifier based on traditional machine learning techniques. The authors in [101] use the K-nearest neighbor method to classify the EMG signals recorded from lower limb muscles during standing in individuals with complete spinal cord injury implanted with spinal cord epidural stimulation. Actually, most KNN-related researches in this area are correlated with other classification algorithms to find the most suitable classifiers for some certain problem. In [102], weighted KNN and SVM are both utilized as the classifier to distinguish the sEMG signals for controlling a prosthetic foot while the comparison of the accuracy of two classifies shows that weighted KNN obtains higher efficiency.

With the rapid development of deep learning technology in recent years, increasing attention has been paid to the deep feature classifier. For example, a backpropagation neural network was used in [109] to map the optimal surface EMG features to the FE angle angles. The experimental results show that the features extracted from multichannel surface EMG signals using deep belief network method proposed in this paper outperform principal components analysis (PCA), and the RMS error between the estimated joint angles and calculated ones during human walking is reduced by about 50%.

## 4. Multisensory Fusion

Through a series of processing, the change of movement patterns creates a stronger correlation with the EMG signal. However, due to the interactions between the lower limb movement and the external environment, as well as the low resolution of the upper limb hand fine movement in the EMG signal, the sensing strategy relying only on the EMG signal has great limitations in these two types of rehabilitation scenes. To address these issues, multisensory fusion techniques are proposed and proved to be capable of boosting pattern recognition accuracy by collecting more efficient data from different dimensions.

Generally, the EMG-centered fusion strategy can be divided into two types, i.e., fusion with kinematic sensors and fusion with kinetic sensors. The former strategy is mainly to supplement the three-dimensional motion information like human body acceleration, angular velocity, angle, etc. While the latter one is mainly to supplement the force information during the movement process. In this section, we will explain the detailed procedures of these two fusion strategies and demonstrate some successful applications using these techniques in different rehabilitation scenarios. Figure 7 demonstrates a block diagram of the above EMG-centered fusion strategy.

### 4.1. Fusion with Kinematic Sensors

Kinematic sensors are widely used in obtaining the locomotion properties of targets. In recent years, kinematic sensors have also attracted enough attention in human behavioral study combined with biosignal sensors (e.g., EMG). Kinematic information of human lower extremity describes the movements of the joints and limbs. Generally, the kinematic sensors can be divided into two main categories, inertial sensors and motion capture systems. Inertial sensors, including accelerometer, gyroscope, and magnetometer, are often used to collect kinematic data like joint angles, joint angular velocity, body orientation, and limb acceleration, etc. Motion capture systems became popular only in recent years, their biggest advantage being that they can collect locomotion information in a contactless manner.

Kinematic sensors’ advantage of intuitively reflecting the 3D motion information of the limb helps compensate for the defect of EMG sensors, which only describes the whole physical process from the human nerve activity angle. Combing bioelectrical signal and motion information like acceleration has also been investigated to improve locomotion classification accuracy. Researchers in [110] especially investigate the improvement in classification error while utilizing EMG and kinematic data together rather than pure kinematic data. The empirical results demonstrate that the higher recognition rates can be obtained with combined data compared with vanilla kinematic information from the hip joint of non-dominant/impaired limb and an accelerometer.

The most successful use of the kinematics-EMG fusion strategy lies in the control of powered lower-limb prostheses. Such devices help people get rid of the disability and restore normal walking ability. Nevertheless, the adaptability of the prosthesis to other parts of the human body is still a challenging subject in which the kernel issue can be regarded as inferring human locomotion intent. Due to the restriction of the traditional recognition approach only relying on kinematic data, some recent works are making the steps to fuse EMG and mechanical information to improve the performance. The locomotion recognition system used for lower limb prosthesis control in [35] fuse the multichannel EMG signals and accelerometer measurements in feature level. The physical test shows that the empirical accuracy is over 95%, which is significantly better than traditional schemes, only applying surface Electromyography for identifying locomotion modes [29,35]. In [111], A combination of kinematic and electromyographic (EMG) signals recorded from a person’s proximal humerus was used to evaluate a novel transhumeral prosthesis controller. Most especially, they trained a time-delayed artificial neural network to predict elbow flexion/extension and forearm pronation/supination from six proximal EMG signals, and humeral angular velocity and linear acceleration. Young, A. J. [15] studied the contribution of EMG data in combination with a diverse array of mechanical sensors to locomotion mode intent recognition in transfemoral amputees using powered prostheses. And this can indeed significantly reduce intent recognition errors both for transitions between locomotion modes and steady-state locomotion.

In addition, EMG centered kinematic fusion technique is also seen in applications of the hand gesture recognition. This is because the dexterous movement of the finger movements varies more significantly in the position information than the EMG information. Therefore, it is an ideal choice to combine them to obtain intention information and position information at the same time. For example, inertial measurement unit(IMU) and myoelectric units are utilized to implement the hand gesture-based control of an omnidirectional wheelchair in [112]. The system component, sensor placement, and the characteristic collected signals are demonstrated in Figure 8. In this system, the classification, which involves recognizing the activity pattern based on the periodic shape of triaxial wrist tilt angle and EMG-RMS from the two selected muscles, helps the accuracy improve to 94%.

Utilizing vision technique is another effective and direct way to detect the kinematic information of human. Recently, with the vigorous development of computer vision, some researchers are trying combining the vision and EMG information for limb motion classification task. In a newly published work [113], researches implemented a framework that allows the integration of multi-sensors, EMG and visual information, to perform sensor fusion and to improve the accuracy of hand gesture recognition tasks. For embedded applications, even-based cameras were utilized to run on the limited computational resources of mobile phones. The online results of hand gesture recognition using fusion approach reached 85%, which is 13% and 11% higher than utilizing EMG and vision individually.

### 4.2. Fusion with Kinetic Sensors

Kinetic sensors are a kind of instrument used to measure forces and moments that are directly connected with the movement of body segments like lower limbs. Various types of force transducers, including piezoelectric [114,115], strain gauged [116,117], and capacitive transducers [118,119,120], have been widely used in the design of force-sensitive sensors. Joint use of kinetic sensors and electromyography sensors is believed to be able to extract human muscle reflex related information, which will help the recognition of specific body behaviors. Generally, two types of kinetic information, interaction force, and ground reaction force can both be integrated with EMG information. The following part will introduce the fusion examples with interaction force and ground reaction force, respectively.

Interaction force sensors are typically embedded in a fixed structure and then used to detect the extension or flexion of limb, which are important for interpreting the intention of the human movement. The utilization of interaction force and EMG signals are also seen in the application of the upper limb hand gesture recognition in recent years. The research in [27] showed that combining EMG and pressure data sensed only at the wrist could support the accurate classification of hand gestures. Especially, the EMG is suited to sensing finger movements, the pressure is suited to sensing wrist and forearm rotations, and their combination is significantly more accurate for a range of gestures than either technique alone. Figure 9 demonstrates the implementation of this multisensory approach in hand gesture recognition.

According to our literature review, ground reaction force (GRF) is also a commonly used kinetic information, especially in lower limb rehabilitation scenarios. A typical combination pipeline focus on lower limb prosthesis is shown in Figure 10. Such integration strategy is mainly applied in the lower limb exoskeleton or prosthesis because the lower limb is mostly responsible for interaction with the ground, and many locomotion modes are closely related to the terrain changes, which is directly reflected in the GRF information. Liu, Ming. et al. [35] applied the EMG measurement results and mechanical ground reaction forces/moments recorded using a six-degree-of-freedom load cell to build an adaptive classification strategy, which further enhanced the reliability of neutrally-controlled prosthetic legs. The researchers in [60] presented a smart sensing system utilizing flexible electromyography sensors and ground reaction force sensors for locomotion mode recognition. Here, EMG and GRF information were collected from ten healthy subjects in five common locomotion modes in daily life. Rehabilitation robots is an attractive research zone which mainly helps the patients like stroke regain the locomotion ability. In [121], ground reaction forces and lower extremity electromyography are both utilized to compare inclined treadmill walking and turning conditions with its emulation on a developed balance assessment robot in order to investigate the feasibility of the existing approaches in rehabilitation.

### 4.3. Fusion with Both Kinematics and Kinetic Sensors

In addition to the information fusion work introduced above, we also notice some recent work combing EMG with both kinematic and kinetic data. With the improvement of processor performance and the development of machine learning algorithms to handle high dimensional data, this triple-type sensor information fusion, which used to be regarded as a difficult approach, has been implemented and outperforms existing approaches in some motion classification tasks.

Rehabilitation exoskeleton is an important application of kinematic-kinetic fusion strategy. Because this kind of equipment needs various dimensions of information to maintain the highest stability and safety, it can achieve the goal of using EMG as the subjective control input and kinematic and dynamic data as the feedback. The work in [122] used human-robot interaction force, ground reaction force, and EMG sensors to distinguish the walking environment and gait period to provide crucial information for exoskeleton control. The results obtained using an individual mechanical sensor together with sEMG showed improvement compared to the case of only using force sensors, reaching the highest accuracy of 97.8% at gait phase recognition scenarios. The sensing blocks, as well as the location of the sensors, have been demonstrated in Figure 11.

Gait pattern analysis has also been widely used to assist in the diagnosis of muscle diseases. By multi-sensor data fusion, gait analysis gives objective and quantitative information about the gait pattern and the deviation due to the muscular situation of these patients, which are of great significance to the clinical research. Cui, Chengkun, et al. [121] conducted the simultaneous recognition and assessment of post-stroke hemiparetic gait by fusing kinematic, kinetic, and electrophysiological data. In [123], researchers investigated the gait pattern of 10 patients with myotonic dystrophy (Steinert disease) compared to 20 healthy controls through manual muscle test and gait analysis in terms of kinematic, kinetic and EMG data.

Table 2 summarizes the fusion strategy, targeted recognition classes, extracted features, proposed classifiers, and the final accuracy results of the multisensory application in different scenarios of rehabilitation.

## 5. Challenges and Future Development

Currently, fruitful results have been achieved by EMG-centered multisensory pattern recognition methods in the rehabilitation field. However, there are still limitations in terms of accuracy, robustness, data volume, and flexibility when current approaches are applied to practical applications. In this section, we will explain the main issues and potential solutions and research directions of multisensory EMG PR.

### 5.1. Low Data Quality

Low data quality is a pressing problem in EMG detection, especially under a multisensory fusion scenario. Because the data quality of each dimension can significantly affect the final recognition accuracy. Since bioelectrical signals are generally more susceptible to noise interference, in EMG centered multisensory fusion, the data quality of EMG signals is the most important influential factor.

In Section 3, we explain the preprocessing of the EMG signal, whose purpose is mainly to reduce the noise in the raw EMG signal. Although by the preprocessing technique, the deterministic noise, e.g., charger noise, can be filtered and stochastic noise such as white noise can be compressed, interference comes from the source of the bioelectric signal that is challenging to be removed by data processing technique. First, the EMG crosstalk effect is an inevitable interference in EMG detection [22]. It means that the amplitude of the EMG signal is not only contributed by the targeted muscle but also influenced by the adjacent and surrounding muscles, hindering precise detection of the muscle activity. Second, electrode shift also results in severe EMG signal misregistration, since the traditional electrochemical gel electrode suffers difficulty in firmly attaching to the skin; a shift of merely 1 cm would reduce the classification accuracy by 5% to 20% [125]. Third, even when the patient performs the same locomotion under the same environment, the applied muscle force may slightly change due to the impossibility in perfectly controlling the body, resulting in the variations of the EMG signals of the selected muscles, and reducing the data quality [19].

Advanced sensing and detecting means could be possible solutions to the problem of low-quality data. With the integration and miniaturization of sensory circuits, high-density EMG (HD-EMG) has become a promising sensing approach. This type of sensing technique increases the coverage and density of the electrodes. Hundreds of electrodes are integrated with a small size overlying a restricted area of the skin [126,127]. The most significant benefit of HD-EMG is that the possibility of subtracting information at the motor unit level greatly improved compared with conventional EMG, minimizing the effect of EMG crosstalk at this precision level. The multiple electrodes are arranged in the form of a two-dimensional array, for example, in the shape of 8 × 24 [127] or 8 × 16 [126], which will produce an “EMG image”. The EMG image presents a comprehensive view of EMG signals both in the time domain and spatial domain, providing far richer information than the conventional EMG electrode placement method of implementing one or two electrodes at a single muscle. For example, the HD-EMG arrays attached to the elbow area were used for the precise hand gesture recognition, reaching the accuracy as high as 99.0% in classifying twenty-seven gestures [126]. A typical procedure of utilizing the “EMG image” generated by the HD-EMG to conduct pattern recognition is demonstrated in Figure 12. Even though abundant information can be extracted with the help of HD-EMG, it also increased the burden of the acquisition and processing hardware. Because the sampling rate of EMG signals requires at least 1000 Hz at each channel, an 128 channel HD-EMG system will produce over 128,000 observations in one second, which would make the data scale enormous and lead to challenges in real-time processing.

### 5.2. Inadequate and Undisclosed Data

High-quality data acquisition is the premise of high-precision multisensory pattern recognition. Meanwhile, for multi-sensor fusion, due to the large variability in the patient’s muscle and physical condition and the difficulty in obtaining high-quality multisensory data, adequate and open-source data is of equal importance. This section will explain the necessity of establishing a big and open sensor data library in detail, and review some currently developing databases.

Currently, most of the current EMG researches were recruiting a relatively small volume of the volunteers to collect EMG signals, such as 5–20 healthy individuals or no more than five patients. The relatively small size is owing to the limited access to experimental resources due to the insufficient funding budget. Therefore, recruiting tens of participants is a balanced choice for verifying the proposed algorithms for most research groups. Nevertheless, there are a few concerns about this choice.

Firstly, the variation of the EMG signal is usually significant in different individuals. A strong correlation between EMG signals and the age, height, weight, exercise habits, and health status of individuals are found in relevant studies [128]. The sample size of tens of experimenters is often difficult to cover all these differences between various populations. Besides, researchers usually adopt their experimental protocols and different sensing equipment, and it is not easy to investigate the generality of the method proposed by one work across these influential factors.

Furthermore, the requirement of sensing equipment for robust EMG acquisition is quite high, and the collection system is typically expensive. It may potentially hinder some labs or researchers who do not have the experimental conditions from investigating EMG signals on the algorithm level.

An important approach for solving the above problem is to promote the concept of “big data-based EMG,” which means that the verification method of EMG pattern recognition should shift from the verification in small sample experimental data to the performance of the proposed algorithm based on big data in different populations. Well-established datasets not only allows researchers to examine the quality and correctness of their acquired data more conveniently but testing the generality of their proposed methods in different datasets. In addition, larger datasets enable the application of some advanced algorithms such as deep learning in digging the underlying information in EMG signals [129].

In the past, the experiment data of EMG were mostly stored within the laboratories of individual researchers rather than publishing online, which made it hard to gather multiple datasets. In recent years, with the occurrence of “data paper,” researchers are encouraged to publish their experimental data as an individual publication [130]. Such scientific publications promote the transparency of the raw data, and thus more EMG datasets are being available. In addition, there are also studies dedicated to establishing a standard dataset, trying to provide a standard experimental procedure and cover as many different muscles, movements, and population as possible. A few influential datasets such as Surface Electromyography for the Non-Invasive Assessment of Muscles (SENIAM), The Ninapro project [131] have emerged in recent years. However, there is still a long way to go in reaching the consensus of a widely accepted protocol and establishing a thorough dataset.

### 5.3. Discrete Interpretation of Continuous Movements

The development of the data quality and quantity offers the possibility to develop more effective processing techniques, especially in improving the simultaneity and continuity of the EMG-PR-based control strategy, which are two key drawbacks at the state of the art technique. In this section, we will explain how simultaneity and continuity problem is affecting the EMG PR performance, and provide possible solutions and research directions in the light of the multisensory technique.

As mentioned earlier, the EMG PR technique is widely used in the real-time Human-Machine Interface system (HMI). However, the broad use of clinical viable EMG based robots are still not reported yet. The potential problem is that a complete pipeline, including sensing, signal transmission, and processing, has to be done in order to recognize the current pattern. There is a time delay in this procedure, and it is acceptable in some scenarios, such as sign language recognition, when only static patterns like hand gestures are to be recognized. However, the patterns relevant to lower limbs are mostly dynamic and require real-time and instant recognition, and therefore the time delay may greatly influence the application of conventional PR technique in lower limb HMI devices.

There are several attempts to resolve the problem of discrete interpretation. One of them is to design the transition logic between two consecutive patterns skillfully, and some pioneers studies are trying to design the transition strategy. In [42], overlapped windows are utilized to overcome the time delay problem, and the comparison between the overlapped windows and non-overlapping windows are depicted in Figure 13. The increment of two consecutive and the overlapped window is only 12 ms, generating approximate continuous decision flow. The majority vote method is used to prevent misjudging problem. In addition, because the EMG signal is generated before the actual motion, there are also studies using the currently collected signals to predict the patterns in the next moment so that the robotic actuators could react in advance to adapt to the changing patterns [132,133]. However, the prediction method is only applicable in the periodical movement like gait, and still have difficulties in predicting more complicated motions. So far, there is still no consensus on the best transition method among EMG patterns, and the robust transition is a research interest in the EMG PR field.

With the development of multisensory fusion and the increasingly thorough study of human body motion, the method of combining the biomechanical model with pattern recognition is also attracting increasing attention. Some open-source software, like OpenSim, is developed to provide present biomechanical models. With the help of the multisensory fusion, both the EMG signals and other mechanical signals collected from the patient’s body can be used to calibrate and specialize the raw biomechanical models. After that, with proper mapping strategies, these models can be driven by EMG signals and, therefore, intuitively simulate the movement of human extremities, providing crucial information for continuous control of HMI systems. For example, a model-based control strategy is proposed in [134], using the EMG signal as the input for the carefully tuned and calibrated hand-finger biomechanic model, and managed to continuously control a prosthesis hand to finish the task of gripping in the response time of 16.2 ms per loop. The control schemes and multisensory strategy is demonstrated in Figure 14. Nevertheless, the model-based method requires tedious procedures to calibrate and tune the model, and it is proved difficult to separate the EMG signals into distinct functional movements for the subject with small arms. At present, it is still in the laboratory research stage.

### 5.4. Future Analysis

In the above three sections, we explain the challenges of multisensory EMG PR and the emerging technologies potentially capable to solve these challenges. We believe that in the foreseeable future, multisensory EMG PR has high possibility to develop in these three directions for the following reasons. Firstly, the application of the novel EMG sensing technique, e.g., High-Density EMG, will greatly improve the EMG signal’s quality. Besides, the spatial resolution of the EMG signals will also be enhanced from the level of distinguishing different muscle tissues to the level of distinguishing single cells, making it possible for EMG signals to apply in detection of very fine movements. Secondly, more open source databases with high data quality, complete muscle measurement points and rich application scenarios will be built, which will further reduce the obstacles of self-designed experiments and attract more researchers to join the field of EMG signal analysis. Thirdly, with the reduction of EMG research barriers, the research of EMG analysis algorithm will continue to increase, and more advanced algorithms to solve the traditional PR problem, such as the overlapping window and biomechanical model mentioned in this paper, will be proposed and verified.

## 6. Conclusions

Employing electromyography (EMG) centered multisensory integration pattern recognition (PR) techniques demonstrates strong potential in precisely interpreting patients’ intentions and locomotion modes, which are of significance for analyzing the progress of rehabilitation status. However, commercialized products widely applied in hospitals or home-based internet of health things (IoHT) have not been reported yet. This article aims to reveal the reason behind this phenomenon by thoroughly reviewing the state-of-the-art research outcomes and providing a comprehensive analysis of limitations. The content provided here not only allows readers to have a full picture view of this area but also shows potential directions in further enhancing the development of applying EMG centered multisensory fusion PR techniques in rehabilitation associated applications.

## Figures and Tables

**Figure 1 biosensors-10-00085-f001:**
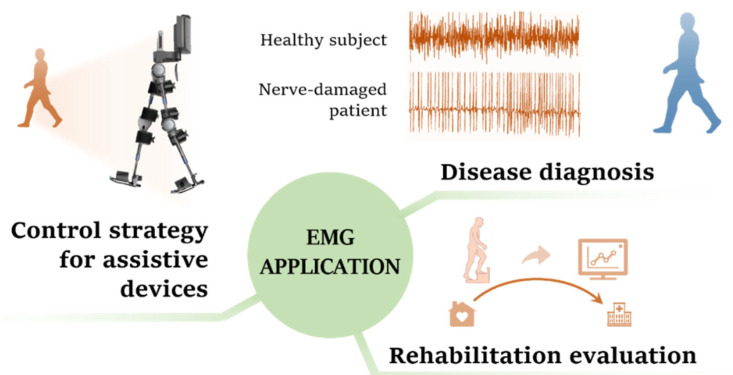
The conceptual depiction of the application scenarios of the electromyography (EMG) pattern recognition technique.

**Figure 2 biosensors-10-00085-f002:**
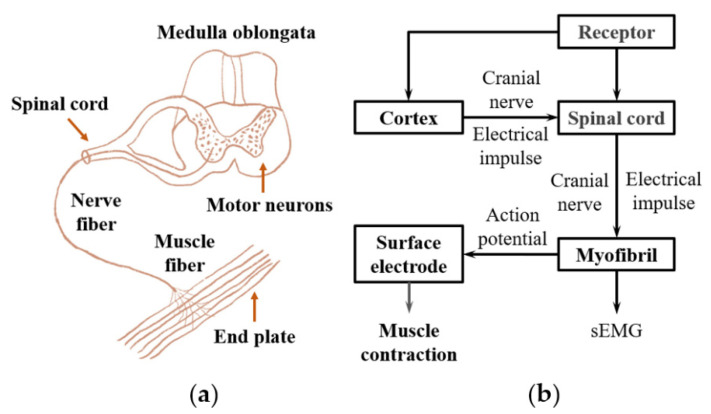
The principle of generation of the EMG signal. (**a**) the structure of the neuro-muscular system. (**b**) the schematic of the EMG signal transduction in the nerve and muscle system.

**Figure 3 biosensors-10-00085-f003:**
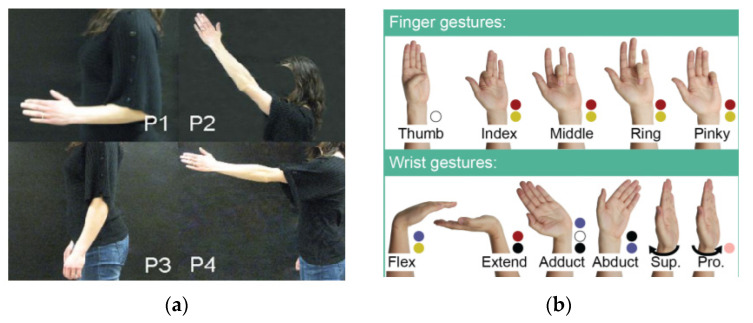
Two types of upper limb patterns. (**a**) the limb position, adopted from [25]. (**b**) hand gestures, adopted from [27].

**Figure 4 biosensors-10-00085-f004:**
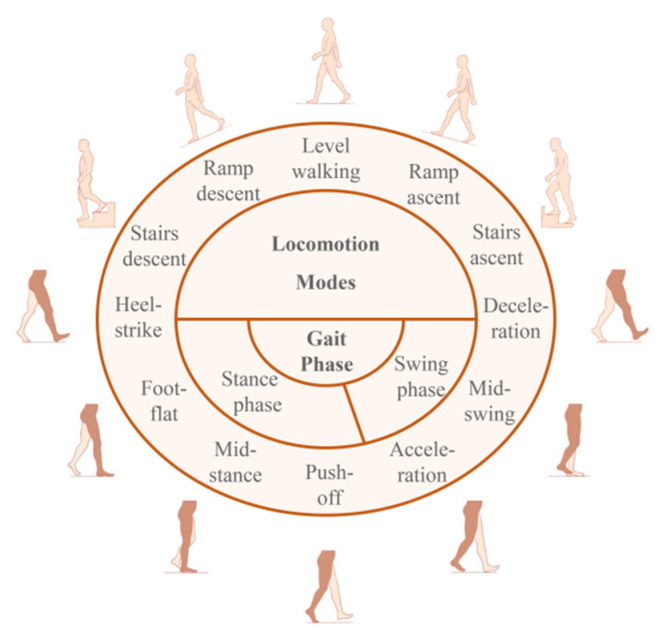
The two commonly studied lower-limb patterns, i.e., locomotion modes and gait phase. Five locomotion modes, along with the definition of the eight-phases gait cycle, are presented here.

**Figure 5 biosensors-10-00085-f005:**
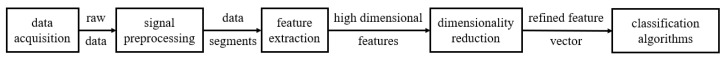
The pipeline of performing EMG pattern recognition.

**Figure 6 biosensors-10-00085-f006:**
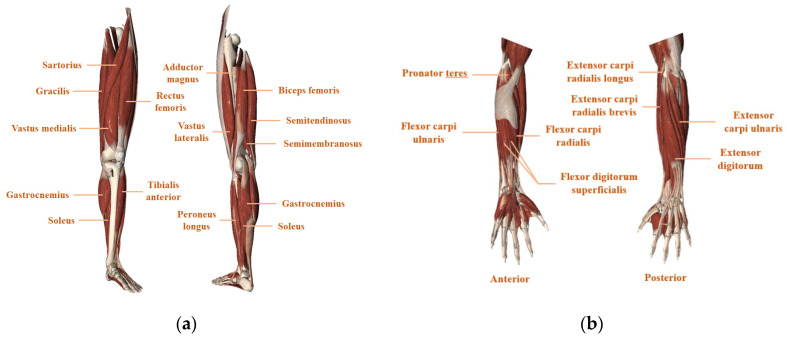
Longitudinal representation of the muscle groups of (**a**) the lower limb (shank and thigh). (**b**) the upper limb (elbow and hand).

**Figure 7 biosensors-10-00085-f007:**
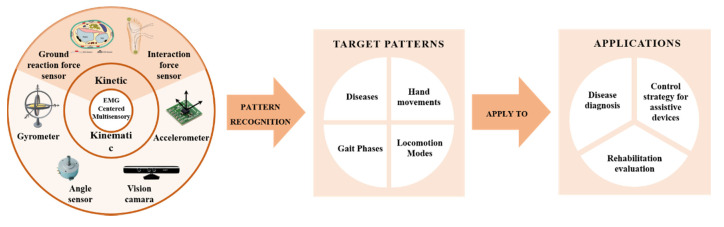
EMG-centered multisensory strategies and sensing blocks.

**Figure 8 biosensors-10-00085-f008:**
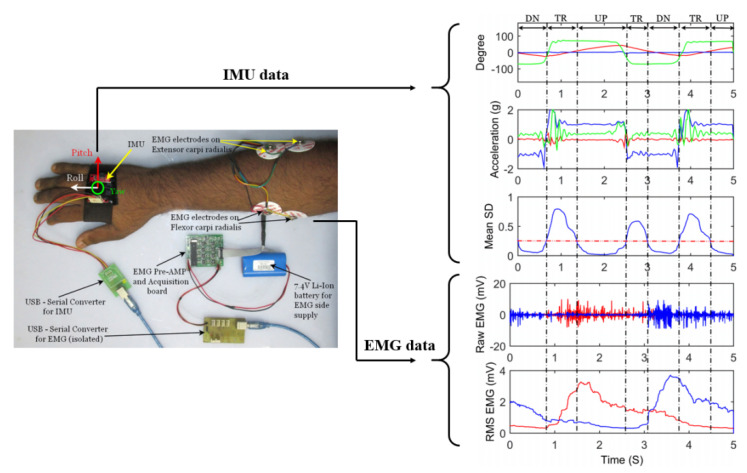
The fusion strategy of combining kinematics data (IMU) and EMG data, adopted from [112].

**Figure 9 biosensors-10-00085-f009:**
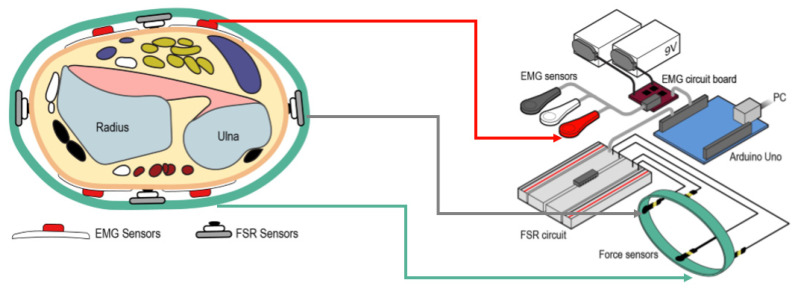
EMG-FSR fusion strategy and hardware diagram used to classify different hand gestures, adopted from [27].

**Figure 10 biosensors-10-00085-f010:**
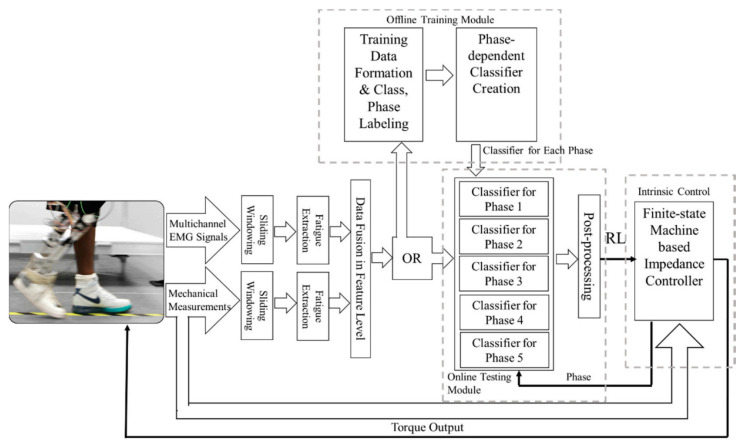
EMG-GRF fusion strategy and the fusion pipeline to recognize different locomotion modes adapted from [27].

**Figure 11 biosensors-10-00085-f011:**
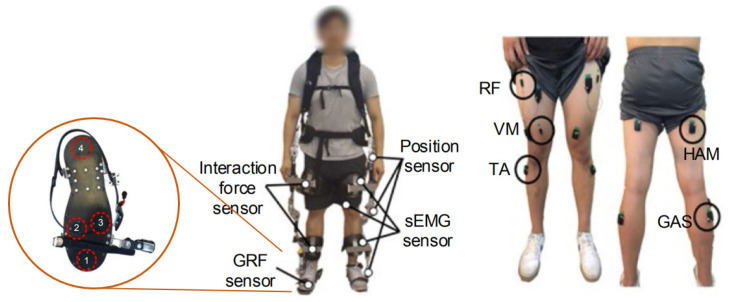
Multisensory fusion strategy using EMG, kinematic sensors, and kinetic sensors together for the application of rehabilitation exoskeleton, adopted from [122].

**Figure 12 biosensors-10-00085-f012:**
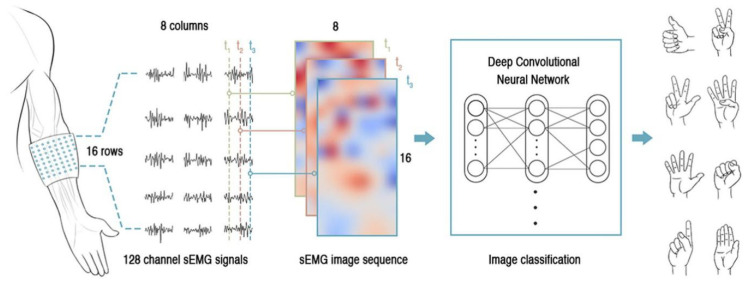
The procedure of applying High-Density EMG (HD-EMG) in discriminating hand gestures, adopted from [126].

**Figure 13 biosensors-10-00085-f013:**
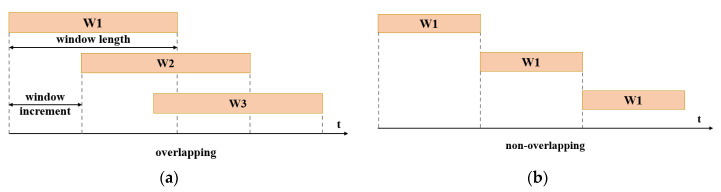
A conceptual depiction of: (**a**) the overlapping windowing method. (**b**) the non-overlapping windowing method.

**Figure 14 biosensors-10-00085-f014:**
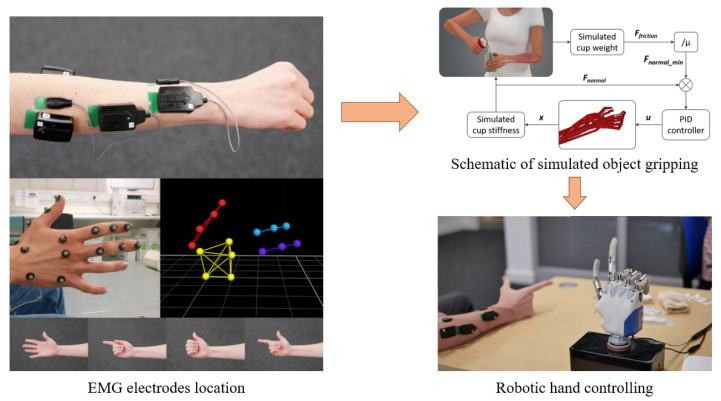
The schematic of utilizing biomechanical based and EMG centered multisensory technique in controlling a robotic hand, adopted from [134].

**Table 1 biosensors-10-00085-t001:** Summary of the selected muscle group for detecting electromyography (EMG) signals.

No.	Proximal Muscle	Distal Muscle	Comments
[42]	SAR, RF, VL, VM, GRA, BFL, SEM, BFS, ADM	/	Gluteal muscles (gluteus maximus and gluteus medius) on the amputated side and the thigh muscles of the residual limb were monitored
[35]	RF, VL, VM, BFL, SEM, BTS, TFL	/	The accurate electrodes locations are adjusted according to the able-bodied subjects and transfemoral subjects
[29]	SAR, RF, VL, VM, GRA, BFL, SEM, BFS, ADM	/	It should be noted that the locations of EMG electrodes on the distal muscles were approximate
[60]	/	TA, SL	Although only two muscles are selected, the classification accuracy is still satisfying
[59]	TPA, DPA, PMC, BCL, TBL, FCR, ECR	/	One of the eight signal channels is used for the synchronization of data from the Fastrack while the left seven are utilized to collect muscle activities signal.
[27]	/	FDS, FDP, EDC, EIP, EMP	These selected muscles are responsible for controlling all fingers except the thumb.
[9]	AM, GM, PRF, VL, VM	/	The proximal hip muscle groups have higher rates of the change in EMG activation with regard to different walking speeds while the distal knee extensor muscle groups show higher rates of change for different waling slopes
[61]	GM, RF, VL, BFL	TA, GA, SL	Humans often change gait patterns to prevent overexertion and possible injury to the relatively small dorsiflexor muscles, which are walking close to maximum capacity.
[63]	RF, VL, SEM		These three thigh muscles are the most commonly used muscles to classify locomotion modes at different speeds.
[64]	BF, RF	MG, TA	To reflect the effect of gait speed and gender on joint motion of lower extremity more comprehensively, bilateral lumbar erectors spinae are also utilized besides the muscles mentioned before.

Muscles: TA = Tibialis Anterior; SL = Soleus; SAR = Sartorius; RF = rectus femoris; VL = vastus lateralis, VM = vastus medialis; GRA = gracilis; BFL = biceps femoris long head; SEM = semitendinosus; BFS = biceps femoris short head; ADM = adductor magnus.

**Table 2 biosensors-10-00085-t002:** Summary of the pattern recognition techniques applied in relevant studies.

No.	Applied Sensors	Classes	Feature	Classifier	Accuracy
[35]	EMG + GRF	Five common locomotion modes (W, RA, RD, SA, SD) and eight task transitions: W->SA, W->SD, W->RA, W->RD, SA->W, SD->W, RA->W and RD->W	EMG data feature: MAV, SL, SSC and ZC, mechanical signals: maximum, minimum, mean value and standard deviation	Entropy-based adaptation (EBA), Learning form testing data (LIFT) and Transductive Support Vector Machine (TSVM)	EBA: 95%, LIFT: 95% and TSVM: 96.25%, vanilla SVM: 87.5%
[42]	EMG + GRF	Locomotion modes: LW, SO, SA, SD, RA and RD and related transitions: W->sA, W->RA, W->O, SD->W, RD->W, SA/RA->W, W->SD/RD	EMG time-domain feature: MAV, SSC, WL, ZC, Mechanical signal features: maximum, minimum, mean value of each direction of force and moment	SVM	99% or higher accuracy in the stance phase and 95% accuracy in the swing phase
[122]	Position sensors, GRF, interaction force EMG	Five walking environments: LW, RA, RD, SA, and SD Seven gait periods: LS, MST, TST, PS, IS, MS and TS	GRF feature: four positions in the foot for four time periods, position feature: three joint angles for four time periods. Interaction force feature: two points in the link for four time periods, sEMG feature: MAV, ZC, SSC and WL	BLDA	96.1%(environment classification accuracy) 97.8%(gait period classification period)
[27]	EMG sensor, pressure force sensor	Finger gestures, wrist gestures, and other gestures	Root mean square (RMS), standard deviation (SD) and peak amplitude	SVM	95.8%
[112]	IMU, EMG sensor	Six hand gestures (forward, clockwise, left, backward, anticlockwise, right)	Nine IMU features extracted from wrist Euler angle and six EMG features extracted from EMG RMS signal	DSVM	Real-time recognition accuracy 90.5%
[124]	EMG signal acquisition system, data glove	Thumb flexion, finger flexion, thumb opposition, middle/ring/little finger flexion, long fingers flexion, tradigital grasp, lateral grip/key grip	MAV (mean of absolute value)	Locally weighted learning	79% for amputee and 89% for non-disabled participants

Classes: LW = level-ground walking; RA = ramp ascent; RD = ramp descent; SA = stairs ascent; SD = stairs descent; SO = stepping over an obstacle; IT = ipsilateral turning; CT = contralateral turning; SS = standing still; LP = loading response; MST = midstance; TST = terminal stance; PS = pre-swing; IS = initial swing; MS = mid-swing; TS = terminal swing. Feature: MAV = mean absolute value; ZC = zero crossing; SSC = slope sign changes; WL = waveform length; SL = signal length.

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
