# Peer review of "EMG-Centered Multisensory Based Technologies for Pattern Recognition in Rehabilitation: State of the Art and Challenges"

_biosensors, 2020, doi:10.3390/bios10080085_

Round 1
Reviewer 1 Report
Authors in this submission presented a state of the art research in the area of pattern recognition using electromyography (EMG) signals based on multisensory technologies.
They have reviewed and presented a rather complete collection of published research in this area, they have also included major papers during the last 10 years as these have published from other researchers and research groups.
The structure and the classification of the works they considered in this submission are organized into different sub-topics in a useful and accurate diagram.
A point that should be noted and should be included in more clear and quantified way to a revised version of their submission is the state of art and latest achievements of the last 2 years.
English writing should also be checked in a revised version of this submission.
Reviewer 2 Report
-please add block diagram of the proposed research
-please add photo of application of the proposed research
-please add some sentences about future analysis;
-please cite references from Web of Science 2018-2020 show new knowledge
-It is good idea to compare your approach with other biomedical methods
or show application:
1) Grasp Posture Control of Wearable Extra Robotic
Fingers with Flex Sensors Based on Neural Network ;;;;
2) Pattern Recognition of Single-Channel sEMG Signal
Using PCA and ANN Method to Classify Nine Hand Movements ;;;;;
3) Recognition of Sedentary Behavior by Machine
Learning Analysis of Wearable Sensors during
Activities of Daily Living for Telemedical Assessment
of Cardiovascular Risk
Round 2
Reviewer 2 Report
the paper is good enough to publish